# Identification of Influential Factors in the Adoption of Irrigation Technologies through Neural Network Analysis: A Case Study with Oil Palm Growers

**Diana Martínez-Arteaga** [1,2,*], **Nolver Atanacio Arias Arias** [1], **Aquiles E. Darghan** [3] **and Dursun Barrios** [2]

[1]    Colombian Oil Palm Research Center-Cenipalma, Bogotá 11121, Colombia; narias@cenipalma.org
[2]    Biogenesis Research Group, Department of Agriculture and Rural Development,
       Faculty of Agricultural Sciences, Universidad Nacional de Colombia, Bogotá 11132, Colombia;
       dbarrio@unal.edu.co
[3]    Department of Agronomy, Faculty of Agricultural Sciences, Universidad Nacional de Colombia,
       Bogotá 11132, Colombia; aqedarghanco@unal.edu.co
*    Correspondence: dmartinezar@unal.edu.co; Tel.: +1-(786)5994082

**Abstract:** Water is one of the most determining factors in obtaining high yields in oil palm crops. However, water scarcity is becoming a challenge for agricultural sustainability. Therefore, when the environmental supply of water is low, it is necessary to provide it to crops with the highest degree of efficiency. However, although irrigation technologies are available, for various reasons farmers continue to use inefficient irrigation systems, which causes resource losses. The objective of this study was to analyze the percentage of adoption of irrigation technologies for water management in oil palm crops and to classify the factors influencing their adoption by producers. The method for the classification of influential factors was based on multiple correspondence analysis and perceptron neural networks. The results showed that fewer than 15% of the producers adopt irrigation technologies, and the factors classified as influential in the adoption decision were the age of the palm growers, the size of the plantation, and the access to extension services. These results are the basis for the formulation of effective and focused extension strategies according to the characteristics of the producers and the local and technological specificity.

**Keywords:** farmers; agriculture extension; irrigation efficiency; perceptron neural networks; technology adoption

## 1. Introduction

The growth expectations of the oil palm agroindustry are among the highest globally [1]. Palm fruit derivatives have huge market potential, as the global demand for vegetable oils is expected to be 70% higher by 2050 [2]. This agroindustry has driven rapid economic growth in several tropical countries and contributed significantly to rural poverty alleviation [3]. However, in some countries around the world, especially in Southeast Asia, the expansion of oil palm cultivation has been associated with deforestation and loss of biodiversity [4]. Consequently, there is a need for sustainable agribusiness in all countries with oil palm cultivation. To respond to the demand to make palm oil production sustainable, among other aspects, good management of water resources must be considered since in oil palm production water is a determining factor in yield [5]. In addition, water scarcity, attributed in part to climate change and population growth, has become a global challenge which not only threatens food security, but has also caused food losses in recent years [6,7].

The effects of water scarcity and drought are causes for concern since global water demand is estimated to grow by about 20–30% by 2050 in the industrial, household, and agricultural sectors, with agriculture remaining the leading consumer of freshwater [8].

The debate on the allocation of water resources to the different economic sectors is expected to intensify, as will the need to increase water-use efficiency [9]. Water is a limited basic resource that has been a constraining factor for agricultural production worldwide [10]. Therefore, irrigation plays a crucial role in most countries, allowing for water control and distribution to meet agricultural needs since irrigated lands currently account for approximately 40% of food and fiber production [11]. Irrigation practices are considered responsible for reducing the impacts of climate change and variability and positively affecting crop yield and food production [12]. Furthermore, the irrigation of crops is vital for food security and economic development. Technological and technical aspects are vital for water resource management and protection [13].

For instance, irrigation technologies (drips and sprinklers) are a form of agricultural water management [14,15]; as irrigation technologies they could reduce water scarcity by 19% and save 35% of agricultural water [16]. However, irrigation technologies are generally infrequently adopted, and they gain coverage slowly [7,17], especially when a high initial investment is required [18]. Accordingly, only 14% of the global 275 million hectares of irrigated land uses efficient irrigation methods [19].

In this scenario of water scarcity, the cooperation of users is essential to use water resources more efficiently and sustainably due to the relationship between their agricultural activities and their environment [20]. Consequently, the adoption of technologies in agriculture has gained considerable attention among researchers and extensionists, who have conducted previous studies to understand the factors influencing the adoption of water-efficient technologies [7]. This study aims to analyze the adoption of efficient irrigation and to classify the factors that influence its adoption by oil palm farmers. The results of this study expand the available conceptual framework, which could help governments and institutions to promote extension and communication strategies based on local and technological specificity, and characteristics of farmers. The rest of the paper is organized as follows: Section 2 considers the theoretical arguments and discusses the current conceptual framework, Section 3 presents the data source and analysis methods, while Section 4 discusses the empirical results and Section 5 concludes the study.

## 2. Conceptual Framework

Technology adoption can be defined as the integration of new technology into existing practices [21]. Technology-adoption behaviors differ vastly and are related to various factors. Among the factors reported, household and plantation characteristics, farmer education, access to extension agents, and cultivation patterns tend to influence the adoption of irrigation systems for rice crops in Ghana [22]. Farmers with more experience, higher levels of education, more secure tenure rights, better access to electricity, more significant institutional presence, and awareness of climate effects are more likely to opt for strategies to avoid water scarcity for rice cultivation [23]. Governments, subsidies, and extension service policies play essential roles in promoting the adoption of modern irrigation technologies in China [24]. Plantation size, together with networks and trust, are important factors that explain the adoption of irrigation technologies by grape farmers in Chile [20].

In addition, education, plantation size, field demonstration, training and access to it, water-use associations, neighboring farmers, and subsidy policies significantly improved the adoption of water scarcity technologies in China [6]. Farmers' age, level of education, plantation size, and perceived usefulness of their social networks positively impacted the adoption of irrigation technologies for tomato crops in China [25]. Finally, technology adoption and intensity significantly depend on factors such as the age of the household head, gender, level of education, household size, access to extension services, and the wealth status of small households in Nigeria [26].

This study incorporates variables that are considered influential, predictable, and consistent in technology adoption (Figure 1).

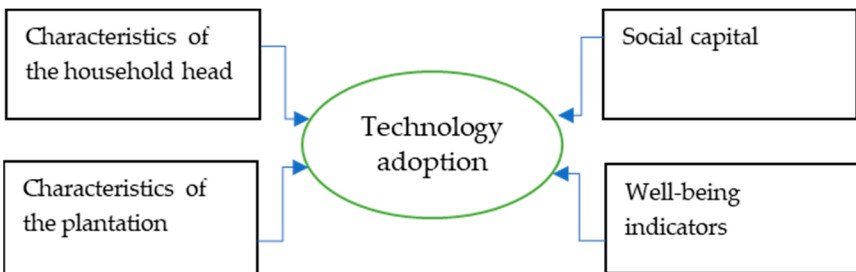

**Figure 1.** Conceptual models of variables that are considered influential in the adoption of irrigation technologies.

The characteristics of the household head include the farmer's age, level of education, and gender. It has been widely established that age has a negative impact on the adoption of irrigation technologies, as the probability of adoption decreases with increasing age [27]. Older farmers are more willing to trust and prefer traditional agricultural practices, and therefore less likely to adopt new technologies [28]. That younger farmers are more innovative and willing to try new developments [25]. Furthermore, those with higher levels of education are more likely to adopt technologies [24] associated with information-gain and farmers' cognitive levels [29], which is consistent with the notion that education helps farmers decide on the adoption of new technologies [22]. Therefore, farmers with higher education and interest tend to understand better and appreciate the advantages of adopting irrigation technologies [6]. Another household-related characteristic that either drives or constrains technology adoption is the farmer's gender. Gender can be related to the characteristics of the technology and the differences and roles among the farming population [26].

Plantation characteristics, including plantation size, are generally associated with wealth and are frequently used in technology-adoption models to capture the impact of wealth on the decision-making process [22]. They suggest that more extensive plantations are more likely to adopt new technologies than smaller plantations [27]. Credit restrictions and access to capital can cause the reciprocal relationship between technology adoption and plantation size. This relationship, in turn, relates to the economies of scale of an investment, whereby more extensive plantations may have more capital and are thus more likely to adopt new technologies. However, all types of farmers may want to do so [20]. Finally, that technology could be more beneficial for farmers with more extensive plantations, making them amenable to adopting new technologies [25]. However, studies on technology adoption indicate no significant association of plantation size with technology-adoption constraints [6]. Another characteristic of the plantation includes whether the plantation is exclusively for productive purposes and whether the farmer has permanent residence, that is, they live on the land where they grow their crops. The close relationship between agricultural activities and the environment that surrounds them can favor the adoption of technologies due to the motivation of having their crops nearby, the need to be efficient with water resources, or the desire to enforce their rights over their homes [26].

Another factor related to technology adoption is social capital. It is defined as an aggregate of actual or potential resources to acquire information [30], share ideas, and gain knowledge. It is characterized by networks, norms, and trust that facilitate the cooperation and coordination of individuals to achieve desired goals and mutual benefit [31]. Thus, belonging to a social network constitutes a means through which farmers can obtain information on new technologies since it creates opportunities for information exchange that lead to a better understanding of the benefits of optimizing irrigation efficiency [27]. Therefore, the participation of farmers in organizations is a critical factor that influences the adoption of irrigation technologies [32]. However, the social effects can be adverse when the networks are vast due to delays caused by the lack of adoption, even when the new technology is more profitable [33]. In addition, globally, agricultural extension tends to be an important source of information on technological improvements in the

sector, especially when the technologies involve different practices [22]. Extension workers effectively disseminate new technologies, increase plantation productivity, and alleviate rural poverty [34]. Similarly, extension services help farmers evaluate costs and determine the risk profile and profitability of innovations [35]. However, the information provided by extension agents regarding the expected return of technology can often be limited. Still, they are an essential source of information on how and when to use new technology [22]. There are also differences between access to technical information and the level of agricultural technology adoption [27].

Finally, well-being indicators are used to judge whether individuals have the basic or optimal inputs for themselves and their family's social and physical development. These conditions involve access to health services, job formality, and infrastructure. This last input (infrastructure) could be one of the most restrictive for adopting sophisticated technologies that require compatibility with plantation conditions. Moreover, it is difficult for farmers to invest in technological innovations that improve income if they face liquidity constraints and have no economic incentives, thus affecting technology adoption [36]. Farmers with liquidity constraints find it challenging to adopt irrigation technologies and are less willing to change due to the risks of possible low crop yields [37]. In addition, since financial opportunities are more limited for economically constrained farmers [32], there is a mismatch between available agricultural technologies and farmers' socioeconomic circumstances [35].

## 3. Data Source and Analysis Methods

### 3.1. Data Source

The data used in this study were collected in a census of 110 farmers (area of 3200 ha) located in the Sevilla River basin in the Magdalena Department in Colombia (Figure 2). The data source was a field survey through face-to-face interviews with farmers on their social and economic characteristics. The interviews were conducted by the Oil Palm Research Center (Cenipalma) in May and July 2021. The information was registered into a mobile application designed by Cenipalma.

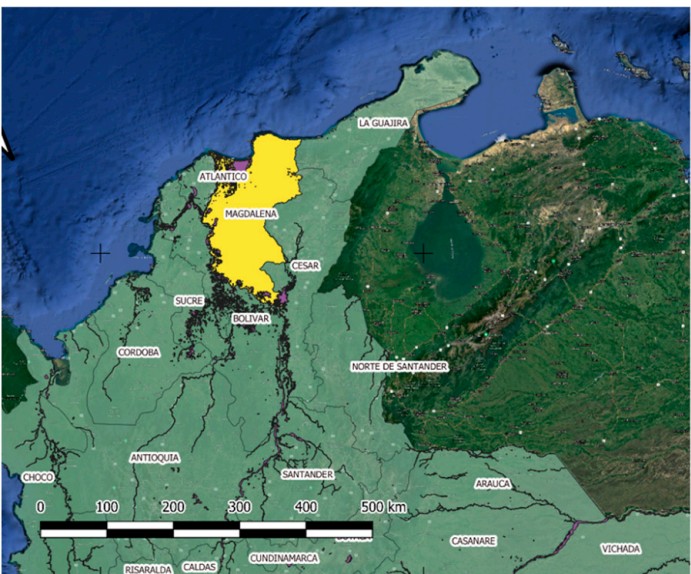

**Figure 2.** Geographical location of the study area.

To meet the objectives of this research, data were collected to analyze the percentage of adoption of irrigation technologies by oil palm producers and the influential variables in the adoption decision (Table 1). The data included the characteristics of the household head—gender, age, and level of education—and the size and use of plantation. For social capital, data included access to extension services and participation in associations or cooperatives.

Finally, the multidimensional poverty index (MPI) was determined to measure well-being indicators. The MPI was based on established socio-economic indicators that are supported and promulgated by internationally accredited institutions (National Council for Economic and Social Policy, 2018). Finally, the drivers implementing irrigation technologies were investigated.

**Table 1.** Definition of the variables of the irrigation technology adoption model.

| Variable | Definition | Mean | S.D. |
|---|---|---|---|
| Age (Ag) | Years | 60 | 15.23 |
| Plantation size (Ps) | Total oil palm area (ha) | 18 | 35.33 |
| | | **Percentage** | |
| Gender (Ge) | (1) Male | 81.8 | 0.39 |
| | (2) Female | 18.2 | |
| Level of education (Le) | (1) Cannot read and write | 3.6 | 0.56 |
| | (2) No education | 53.6 | |
| | (3) Higher education | 42.7 | |
| Use of the plantation (Up) | (1) Production and permanent residence | 12.7 | 0.33 |
| | (2) Exclusively for production | 87.3 | |
| Extension agencies (Ea) | (1) Without access | 27.3 | 0.44 |
| | (2) With access | 72.7 | |
| Association or cooperative (Ac) | (1) Not a member | 44.5 | 0.50 |
| | (2) Member | 55.5 | |
| Well-being indicators (Wi) (Multidimensional Poverty Index—MPI) | (1) Has MPI | 17.3 | 0.38 |
| | (2) Does not have MPI | 82.7 | |

### 3.2. Analysis Methods

This study was observational. That is, researchers looked at the effect of one or more risk factors or another intervention, such as the adoption of technologies, without attempting to change who was or was not exposed to them. In detail, each subject incorporated or did not incorporate the technologies over time and what was done was to study those factors that could have contributed to the adoption. Thus, the technology adoption analysis considered farmers' responses to the irrigation systems used in their plantations. The coding of the technology adoption (Ta) variable was:

1: The farmer adopts irrigation technologies (drips or sprinklers)

0: The farmer has surface/flood irrigation

For a matrix such as the one available, with predominantly qualitative data, correspondence analysis (MCA) works very well in reducing dimensionality, thus allowing us to extract the components with the highest explained variance. Much of what is obtained in observational studies needs to be screened for noise and redundancy so using this technique allowed to simplify and simultaneously detect patterns in the reduced data set so as not to fall into what is known as "the curse of dimensionality". Undoubtedly, other techniques such as dimensional scaling or factor analysis could also have been applied, but the simplicity of interpretation of the former and the possibility of extracting row and column contributions made it more interesting in this case.

In conforming the data matrix with p nominal or ordinal variables, each nominal variable has Jk levels and the sum of the Jk is equal to J. There are I observations. The I × J indicator matrix is denoted X. Applying correspondence analysis to the indicator matrix yields two sets of factor scores: one for the rows and one for the columns. In general, these factor scores are scaled so that their variance is equal to their corresponding eigenvalue. The grand total of the table is denoted by n, and the first step of the analysis is to calculate the probability matrix Y = n − 1X. We denote r the vector of row totals of Y, with c the vector of column totals, and $D_c$ = diag{c}, $D_r$ = diag{r}. Factor scores are obtained from the singular value decomposition.

$$D_r^{-1/2}(Y - rc')D_c^{-1/2} = PUQ' \qquad (1)$$

where $c'$ represents the transpose of c. The estimates of the factors extracted from (1) for row and column were obtained, respectively, from:

$$F = D_r^{-1/2}PU \text{ and } G = D_c^{-1/2}QU.$$

By using the chi-square distance as a metric from the rows and columns to its own barycenter, we get:

$$d_r = \text{diag}\{FF'\} \text{ and } d_c = \text{diag}\{GG'\}.$$

With these last measurements, other metrics such as square cosines are obtained, which help to locate the importance factors in the row or column variables of the data matrix. Another metric called row or column contribution is also possible, which is key for the selection of variables in the data or in the same selection of data rows, which usually correspond to the study respondents. The table of dimension J × J that was obtained with the data arranged in a binary way (due to the multiplicity of scales in each question of the questionnaire) is denoted with B = X'X and is called the Burt matrix associated with X. The matrix of Burt also plays an important theoretical role because the eigenvalues obtained from this analysis give a better approximation of the inertia (data concentration) explained by the factors. In several publications, the MCA interpretation is based on the proximities between points on a low-dimensional diagram (for simplicity, up to three dimensions). These proximities only make sense between points of the same set, rows, or columns. When two points in the row are close to each other, it is likely to select the same levels of the variables. For the proximity between variables, it is necessary to distinguish that the proximity between levels of different variables means that these levels will tend to appear together in the observations, and, since the levels of the same variable cannot occur simultaneously, another interpretation is necessary. Here the proximity between levels means that the groups of observations associated with these two levels are similar.

In conclusion MCA is a specialized multidimensional method measured on a nominal scale; it enables the classification of objects, providing an in-depth analysis of the studied phenomenon (R was used for the MCA). Thus, MCA was initially used for dimension reduction to identify the factors affecting farmers' technology adoption.

After the informative and discriminating variables were identified, perceptron neural networks (PNNs) were used to determine the factors that most influence the adoption of irrigation technologies. Regarding the neural network, whenever dimensionality is reduced in a study such as this one, and the results are used to classify the respondents or subjects of the study, it is of interest to apply some classification algorithm, and networks have worked well for this purpose, so this technique was used because of its simplicity of use at present, although some other classification algorithm such as cluster analysis could also be applied.

The multilayer perceptron neural network (PNN) is the best known of the neural networks and therefore the most widely used. These types of networks are very powerful and dynamic and can be complex. The use of several layers is attributed to the need to increase the complexity of decision regions, which mathematically translates into half planes or intersections between them. This justifies the use of the PNN to act as a universal approximator. Each of your units yields one weighted sum of its inputs which is added to one, which is passed through the activation function. The model is illustrated in the Figure 3.

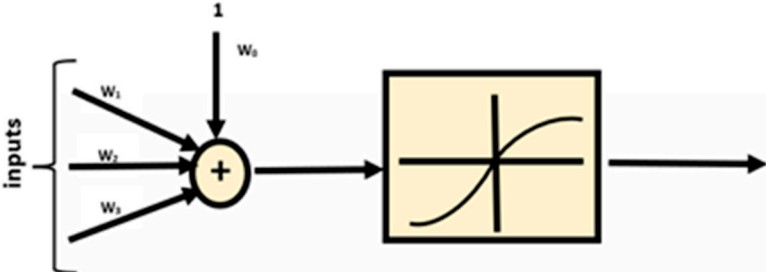

**Figure 3.** Illustration of a multilayer perceptron neural network (PNN).

The number of neurons in the input layer is equal to the number of measurements for the problem (variables) and the number of neurons in the output layer is equal to the number of levels in the classifier (response). The main objective is to optimize the classification task.

Starting from $n_0$ neurons ($x_0, x_1, x_2, \ldots, x_{n0}$) and a specific activation function f, it is necessary to obtain the output of the network in each unit for each layer. For N hidden layers ($h_1, h_2, \ldots, h_N$) and assuming that $n_i$ are the number of neurons for each hidden layer $h_i$, the first output of the hidden layer is given by

$$h_i^j = f\left(\sum_{k=1}^{n_i-1} w_{k,j}^0 x_k\right); \, j = 1, 2, \ldots, \, n_i.$$

where $w_{k,j}^0$ is the weight between the neuron k in the hidden layer 0 and the neuron j.

$$h_i^j = f\left(\sum_{k=1}^{n_i-1} w_{k,j}^{i-1} h_{i-1}^k\right); \, i = 1, \ldots, N; \, j = 1, 2, \ldots, \, n_i.$$

The outputs $h_i^j$ of neurons in the hidden layers are computed as flows: the output of the network once the objective function and the restrictions of existence and communication between neurons have been established can be shown as:

$$Y = \begin{pmatrix} y_1 \\ \vdots \\ y_k \\ \vdots \\ y_{n_{N+1}} \end{pmatrix} = \begin{pmatrix} f\left(\sum_{k=1}^{n_N} w_{k,1}^N h_N^k\right) \\ \vdots \\ f\left(\sum_{k=1}^{n_N} w_{k,j}^N h_N^k\right) \\ \vdots \\ f\left(\sum_{k=1}^{n_N} w_{k,N+1}^N h_N^k\right) \end{pmatrix}$$

Finally, the optimal architecture of the artificial neural network can play an important role in the classification problem for the obtained output [38].

In conclusion, due to the complex environment, dynamic interaction, and data showing nonlinear properties of agricultural systems, PNN allows the analysis of different variables and the classification of the important ones or those with the most significant influence on the observed data [35]. PNN can work with models in several periods and multi-variables without multicollinearity problems or the need to specify the type of functional relationship. PNN can use all the interactions between explanatory variables to estimate the outcome variable better and provide a selection or prediction even when the independent and dependent variables have a nonlinear relationship [39].

The performance and results of the model were analyzed through the ROC (receiver operating characteristic) curve, which visually represents the susceptibility by the specificity of the cut-off points, where a value close to 1 indicates that the model has an excellent ability to classify cut-off points good classification capacity. The network finally used was a multilayer perceptron, trained with the hyperbolic tangent algorithm and the softmax activation function of SPSS version 26.

*3.3. Results of Analysis Methods*

A targeted sample including adopters (14.5%) and nonadopters (85.5%) was used to perform the MCA. A total of 110 rows and 9 columns were analyzed, including the response variable of technology adoption. Based on Burt's tables or generalized contingency tables field [40], the MCA allowed the elimination of categories that did not contribute to total variability. Next, the analysis was performed with a PNN; the partition of 70% of the sample was maintained as an optimal training dataset to learn and find the weights for the neural network structure, 20% for the test sample, and 10% for the reserved sample. This 10% for the reserved sample was used because the algorithm was built with the training and test data. However, it is worth mentioning that the number of adapters available for the development of the model is limited and unbalanced, and there is a risk of poor representativeness since the classification algorithms favor the most frequent class. To relieve this problem, it is recommended to discard some of the variables, even those that contain original data, to avoid excessive training of the neuronal network [41]. For this, the MCA reduced the number of categories to compensate for this problem, leading to the identification of the variables and categories that contribute to total variability. Of the 20 types considered in the survey, 17 were included in the analysis with the PNN to identify the factors that most influenced the adoption of irrigation technologies (Figure 4).

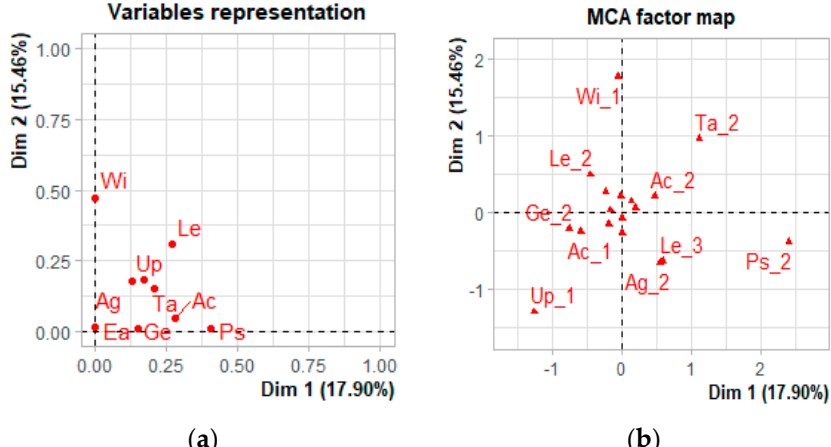

**Figure 4.** (**a**) Graphical representation of the variables of the irrigation technology adoption model and (**b**) representation of the categories that contributed to the total variability in the adoption model of irrigation technologies.

## 4. Results and Discussion

Results for the percentage of adoption of irrigation technologies in the sample showed that 85.5% of farmers had not implemented any technology to efficiently manage water resources, while 14.5% had started sprinkler or drip-irrigation systems. In terms of use of the implementation of these irrigation systems, sprinkler irrigation was the most used.

The second part of the analysis looked for factors that influenced the technology adoption using a neural network approach. Of the eight variables used in the model, three presented significant values—the best values of significance were found in the variables age of the producer, size of the plantation, and access to extension services (Figure 5). It is worth mentioning that the model classified the variables that influenced adoption by the producers; these were consistent with previous work on the adoption of technologies [26,35].

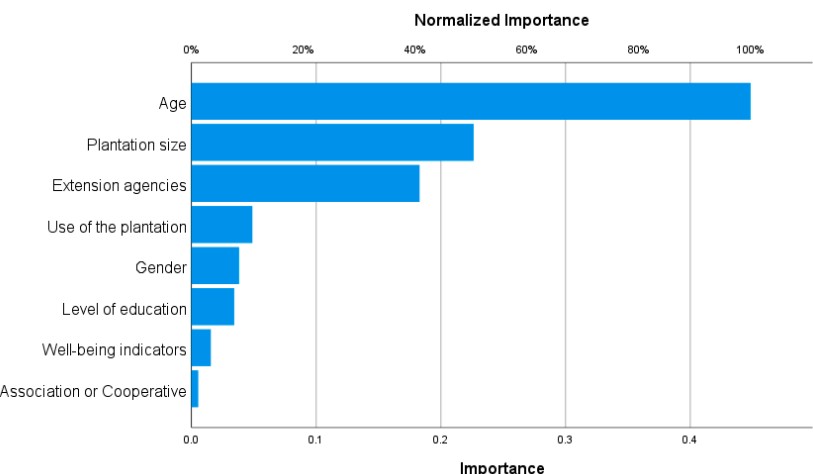

**Figure 5.** Variables influencing the adoption of irrigation technologies in oil palm cultivation.

The age of the producers was the most influential factor in the adoption of irrigation technologies. This result is consistent with other studies on technology adoption which highlight that larger producers are more reluctant to introduce new technologies, since they trust traditional techniques due to the user experience or the zero investment cost. The lack of generational relief was also a factor. There is a high investment required to renew the irrigation technology that saves water resources, and this investment cannot be recovered over time. Older growers are more likely to prefer and rely on traditional techniques [28]. In addition to the above, the lack of knowledge of the impact of the latest technology means that older farmers prefer traditional agricultural practices.

Another influential factor in the adoption of irrigation technologies was the size of the plantation. The smaller the number of hectares, the lower the adoption of complex technologies because the technology can be more beneficial for farmers with more extensive plantations. Higher incomes also facilitate reinvestment in technologies [25]. However, plantations with an average of eight hectares with the adoption of irrigation technologies were found. Therefore, the size of the plantation is a relevant variable and is associated with a greater probability of increasing adoption, but it is not a critical constraint. Additionally, the greater the scarcity of water resources on a farm, the more likely it is that new technologies will be adopted.

Access to agricultural extension, the advisory or support visits that producers receive from the palm union, was another influential factor in adoption. This finding agrees with other related investigations in which a strong relationship was found between extension and the adoption behavior of farmers [20]. One possible explanation is that extension agents are an essential source of information on technological improvements in the sector. In addition, the extension agents exert social pressure and strengthen farmer self-confidence and how they are perceived by other farmers in the community or respective entities [7]. However, only 12% of farmers with access to extension adopted irrigation technologies. This is consistent with the behavioral theory since knowledge can change the intention to adopt technologies, although this is influenced by economic restrictions, access to credit, market incentives, and financing.

Other variables were less influential. The use of the plantation had a marginal effect, although when it was only used for productive purposes, more adoption of technologies was found. In oil palm cultivation, most producers migrate to places with better living conditions and invest in technologies that accentuate the consumption of time, the intensity of labor, and the waste of resources. Consequently, farmers save resources and reduce costs by adopting irrigation technologies.

Sociological factors have a much more significant influence on innovations involving considerable new skills [42]. The level of education had a minor impact. However, it controls farmers' perceptions of whether technologies are beneficial and profitable in a complex environment, where, due to water scarcity, irrigation technologies are a way for the agri-

cultural sector to be efficient and sustainable. Moreover, better-educated farmers have a greater capacity to process information and seek technologies suitable for their production constraints than less educated farmers. The results of this research are consistent with the findings of other studies that show that the farmers' social characteristics are significantly related to technology adoption.

Regarding well-being indicators, farmers lacking primary conditions, such as infrastructure, health, and labor formality, which often represent household wealth, are unlikely to adopt irrigation technologies. Although all farmers want to adopt technologies, owners with plantations with the primary or optimal inputs for social and physical development are more likely to do so. Furthermore, it was observed during field visits that the rural economy and livelihoods were minimal for some farmers.

Associations or cooperatives were of low importance and were related to farmers' participation in the organizations. The results suggest that social effects influence technology adoption. However, they can have adverse effects in the case of large networks where delays occur due to the lack of adoption. Even when the new technology is more profitable, the association may not be strong, or information on the importance of adopting irrigation technologies may not be emphatic or immediate enough. On the contrary, the main drivers for adopting irrigation technologies were water savings, impact on production, and ease of handling the irrigation equipment.

In summary, farmers are aware of the need to optimize their resources while maintaining the highest yields attributable to irrigation. Regarding the critical requirements for implementing irrigation technologies, financing was the most frequent response by the farmers. These results corroborate what was perceived in the field since most farmers questioned the lack of subsidies, access to credit, and support for the commercialization of environmentally sustainable products.

Finally, the area under the ROC curve (Figure 6) was 0.869, a reasonably high value for a multivariate analysis based on nonlinear data of agricultural systems. Therefore, the perceptron network model developed in this study was adequate and allowed for selecting influencing factors in adopting irrigation technologies.

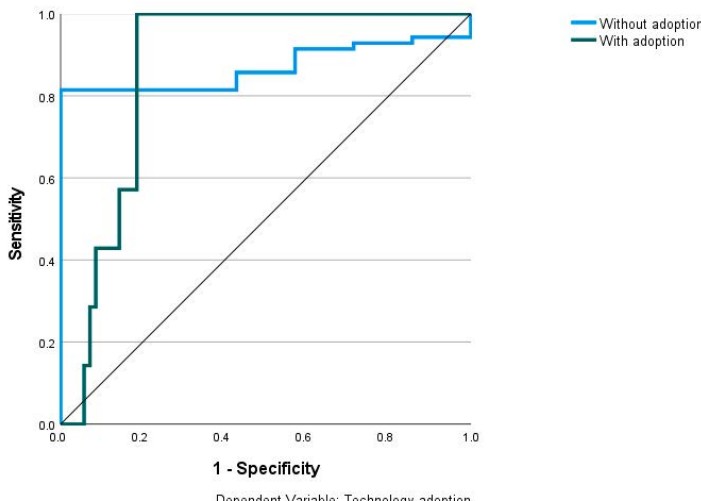

**Figure 6.** ROC curve of neural networks for influencing variables in the adoption of irrigation technologies in oil palm cultivation.

## 5. Conclusions

Factors influencing the adoption of irrigation technologies by oil palm growers were the age of the grower, the size of the plantation, and the access to extension. These results coincide with research findings developed with other statistical techniques and in other agricultural sectors and/or technologies. The other factors—use of the plantation, gender, level of education, indicators of well-being, and participation in associations—had

a marginal effect. Given that the adoption of technologies is a complex process and the influencing variables are not limited to those studied, for future research it is recommended to include, as part of the model, the interaction between companies that provide technical assistance to find the effect that these have on the dissemination and comprehensive support (support for inputs, credits, better payments for the fruit, bonuses for quality or sustainability of the crop, soft technologies, among others) in the adoption of technologies. Finally, the main drivers of adoption were saving water, impact on production, and ease of handling irrigation equipment, while the main impediments were financial limitations.

**Author Contributions:** Conceptualization, D.M.-A., N.A.A.A. and D.B.; data curation, D.M.-A. and A.E.D.; formal analysis, D.M.-A. and A.E.D.; investigation, D.M.-A.; methodology, D.M.-A., N.A.A.A., A.E.D. and D.B.; software, D.M.-A. and A.E.D.; supervision, N.A.A.A., A.E.D. and D.B.; validation, D.M.-A., N.A.A.A., A.E.D. and D.B.; visualization, D.M.-A., N.A.A.A., A.E.D. and D.B.; writing—original draft, D.M.-A.; writing—review and editing, D.M.-A., N.A.A.A., A.E.D. and D.B. All authors have read and agreed to the published version of the manuscript.

**Funding:** The Colombian Oil Palm Research Center-Cenipalma supported this work, and support was also received from the Oil Palm Development Fund.

**Institutional Review Board Statement:** Not applicable.

**Informed Consent Statement:** Informed consent was obtained from all subjects involved in the study.

**Data Availability Statement:** Not applicable.

**Acknowledgments:** This work was supported by the Colombian Oil Palm Research Center-Cenipalma, especially in the extension and geomatics areas. Support was also received from the Oil Palm Promotion Fund administered by Fedepalma.

**Conflicts of Interest:** The authors declare no conflict of interest.

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
