# Peer review of "Identification of Influential Factors in the Adoption of Irrigation Technologies through Neural Network Analysis: A Case Study with Oil Palm Growers"

_agriculture, doi:10.3390/agriculture13040827_

Round 1

Reviewer 1 Report

1. The references in paper should revised with the author's name, please revised them.

2. Figure 1 is missing, please revise them.

3. Why did you use the Multiple Correspondence Analysis and Perceptron Neural Networks? any other methods could be used here?

4. The  principle or models of the MCA and PNN should be described in more detail.

5. What is your experimental scheme? please added them.

6. The influential factors analysis should be presented in more detail.

7. The results analysis should be presented in more detail.

8. The abstracts and conclusions should be improved further.

Author Response

Best regard
I attach a Word document with the respective answers.

Thanks for your time and comments.

Reviewer 2 Report

This is an interesting study on the use of advanced research methodology to identify prime factors on use of irrigation technology (drip and sprinkler) in agriculture. The authors have very clearly highlighted the importance of water in agriculture and the future need for improving the water use efficiency. Improved irrigation technology adoption are very much essential to this direction. The methodology and results are very nicely presented. Age variable being the most influencing factor, may be tested for its confounding effect by the affordability factor. The following suggestions may be incorporated for further improvement of the manuscript.

1. Line 37: Please use a suitable word in place of 'influenced'

2. Line 58: Please give a comma between 19, 20

3. Section 2: Lines 79-175: For easy reading by the readers, the style of writing may be changed with referenes at the end, instead of writing as --emphasized/concluded/noted/showed etc.

4. Fig. 1 is distorted, needs corrections.

5. Line no. 138-139: Please elaborate what is the residential use of oilpalm plantation.

6. A map of the study location may be given.

7. Conclusion part may be made short and crisp.

Author Response

(The authors gave the same response as above.)

Reviewer 3 Report

The adoption and diffusion of different irrigation technologies have captured the attention of numerous researchers in recent decades, there are many existing publications on the matter, due to the nature of the water resource, scarce and necessary for survival. Various methods have been used for its analysis, such as the duration or survival models defined by Jenkin (1995) in which adoption is modeled by the Hazard function. In this paper, the adoption of technology has been studied with a methodology more used in research in other branches of knowledge, such as psychology, which is why if the authors consider it a contribution to their work, they should highlight it, and put in value, as well as accompany it with the mathematical base, or mathematical expressions on which this methodology is based.

Regarding the influence of the age variable in the study, it has been agreed with the majority of the authors that the older the person, the more reluctant the person is to introduce technology, this has also been argued due to the lack of generational change, which It conditions a high investment to renew the production technique towards an irrigation technology that saves water resources and this investment may not be recovered over time.

There are small things to keep in mind:

• In figure 1, it has been out of square

• In figure 2, the term Ac, what does it mean?

• On line 332, what does the acronym RPM mean?

Author Response

(The authors gave the same response as above.)

Round 2

Reviewer 1 Report

The paper was revised well.

Reviewer 3 Report

Thank you for taking the advices, and for substantially improving the work.